# Elucidating the effect of mechanical stretch stress on the mechanism of ligamentum flavum hypertrophy: Development of a novel *in vitro* multi-torsional stretch loading device

**Woo-Keun Kwon[1,2☯], Chang Hwa Ham[1,2☯], Hyuk Choi[3], Seung Min Baek[3], Jae Won Lee[3], Youn-Kwan Park[1], Hong Joo Moon[1], Woong Bae Park[4], Joo Han Kim[1]***

1 Department of Neurosurgery, Korea University Guro Hospital, Korea University College of Medicine, Seoul, Korea, 2 Focused Training Center for Trauma, Korea University Guro Hospital, Korea University College of Medicine, Seoul, Republic of Korea, 3 Department of Medical Sciences, Graduate School of Medicine, Korea University, Seoul, Korea, 4 Department of Neurosurgery, Ewha Womans University Seoul Hospital, Seoul, Korea

☯ These authors contributed equally to this work.
* nskjh94@gmail.com

**Data Availability Statement:** The data obtained from the human participants in this study contain

## Abstract

### Objective

We developed a novel multi-torsional mechanical stretch stress loading device for ligamentum flavum cells and evaluated its influence on the development of ligamentum flavum hypertrophy, a common cause of lumbar spinal canal stenosis.

### Materials and methods

Stretch strength of the device was optimized by applying 5% and 15% MSS loads for 24, 48, and 72 h. A cytotoxicity assay of human ligamentum flavum cells was performed and the results were compared to control (0% stress). Inflammatory markers (interleukin [IL]-6, IL-8), vascular endothelial growth factor [VEGF], and extracellular matrix (ECM)-regulating cytokines (matrix metalloproteinase [MMP]-1, MMP-3 and MMP-9, and tissue inhibitor of metalloproteinase [TIMP]-1 and TIMP-2) were quantified via enzyme-linked immunosorbent assay.

### Results

Using our multi-torsional mechanical stretch stress loading device, 5% stress for 24 hour was optimal for ligamentum flavum cells. Under this condition, the IL-6 and IL-8 levels, VEGF level, and MMP-1, MMP-3, and TIMP-2 were significantly increased, compared to the control.

### Conclusion

Using the novel multi-torsional mechanical stretch stress loading device we confirmed that, mechanical stress enhances the production of inflammatory cytokines and angiogenic

personal health information (i.e. spinal stenosis). The local IRB (http://irb.kumc.or.kr) has not given the permission to export the data externally. You may also contact the IRB under (+82) 02-2626-1868, kughirb@naver.com.

**Funding:** This research was supported by Basic Science Research Program through the National Research Foundation of Korea funded by the Ministry of Education (NRF-2017R1D1A1B03029729) and received by KJH. https://www.nrf.re.kr/eng/main/ The funders had no role in study design, data collection and analysis, decision to publish, or preparation of the manuscript.

**Competing interests:** The authors have declared that no competing interests exist.

factors, and altered the expression of ECM-regulating enzymes, possibly triggering ligamentum flavum hypertrophy.

## Introduction

Up to 70% of the general population experiences chronic lower back pain (LBP) once or more throughout their lifetime [1]. Among the many possible causes of chronic LBP, the prevalence of lumbar spinal canal stenosis (LSCS) in the elderly population is gradually increasing [2]. LSCS is also associated with lower-extremity radiculopathy and neurogenic claudication, which greatly affects the walking distance of the elderly. These clinical symptoms are associated with daily quality of life and therefore are of great interest to spinal physicians. The patho-mechanism of LSCS is unclear, but facet joint enlargement, central intervertebral disc bulging, and ligamentum flavum hypertrophy (LFH) are contributing factors [3]. Among them, LFH secondary to the aging process or mechanical stimulation induced by instability of the spinal segment are key [4]. Therefore, research on the physiologic basis of LFH has caught the attention of spinal specialists, who agree that inflammation, angiogenesis, and matrix regulation of ligamentum flavum influence the development of LFH [5–9].

Mechanical stretch stress (MSS) on the ligamentum flavum is a major contributing factor to LFH. Hayashi *et al.* reported that mechanical stress concentration was directly linked to LFH in a rabbit model [10, 11], and Hur *et al.* emphasized the link between angiogenesis and mechanical stress-induced LFH [7]. Other studies have revealed an association between inflammation triggered by mechanical stress and LFH [6–9]. Nonetheless, it is doubtful whether these studies mimic *in vivo* mechanical stress. In this study, we developed a novel multi-torsional cell plate stretch device that mimics *in vivo* mechanical stress on ligamentum flavum tissue. We evaluated the molecular biological responses related to inflammation, angiogenesis, and extracellular matrix (ECM) regulation that were reported in previous studies of ligamentum flavum cells to various stress loads to identify the stress load that best mimics LFH.

## Materials and methods

### Ethic declaration

This study was reviewed and approved by the local ethics committee (Research Ethics Committee of Korea University Guro Hospital: approval number 2017GR0175) and has been performed in accordance with the ethical standards as laid down in the 1964 Declaration of Helsinki and its later amendments. Informed consent was obtained from all participants.

This study was approved by the Institutional Review Board (IRB) of our institute. Human LF tissues were collected during surgeries on the lumbar spine for herniated nucleus pulposus, following the regulations of the IRB. LF cells were isolated from the tissues of five patients of normal LF thickness. LF tissues harvested in the operating room were placed in sterile Ham's F-12 medium (Gibco-BRL, Grand Island, NY) containing 1% penicillin/streptomycin (P/S; Gibco-BRL) and 5% fetal bovine serum (FBS; Gibco-BRL). After a phosphate-buffered saline (PBS; Welgene, Gyeongsan-si, Gyeongsangbuk-do, Korea) wash, tissues were minced and digested for 1 h at 37˚C in Dulbecco's modified Eagle's medium (DMEM; Welgene, Gyeongsan-si, Gyeongsangbuk-do, Korea) with 0.2% pronase (Calbiochem, La Jolla, CA). Next, LF tissues were incubated overnight at the same temperature in 0.025% collagenase I (Roche Diagnostics, Mannheim, Germany). LF cells were filtered through a sterile nylon-mesh cell

strainer (pore size, 70 μm), centrifuged, and the pellets were resuspended and cultured in DMEM containing 10% FBS and 1% P/S in a humidified atmosphere of 5% $CO_2$ at 37˚C. LF cultures were continued until reaching full confluence. The cells were trypsinized and replated for subculture. Subsequent experiments were conducted using these second-passage LF cells. The detailed LF isolation and culture protocols are adopted from previously reported LF experimental studies [5, 8].

## Design and implementation of the novel MSS loading system

We fabricated a multiple-multidirectional mechanical stretch stress (MSS) loading chamber system capable of incubating dishes containing LF cells. The multi-torsional cell plate stretch device comprises a roofless metal frame containing fixation panels, twisting parts, culture chambers, and a controller. Multiple chambers are seated parallel on the fixation panel facing upwards (Fig 1). The sides of the chamber are fixated to two separate and parallel-oriented fixation panels, which pulls the chamber by moving in the opposite direction. In addition, the fixation panels are coupled to the twisting part to produce torsion stress on multiple chambers. The parallel chambers are aligned and stretched in the same direction and with identical power simultaneously. Each chamber is made of flexible polydimethylsiloxane (PDMS) by photolithography, that can contain cell cultures and stretch or twist. An optically transparent, ultrathin (100 μm) membrane was applied to the well bottom to render the stretch chambers compatible with optical and fluorescence microscopy (Fig 2). The MSS force developed by two step motor generators were controlled by Arduino Uno and L293D motor drivers, regulating the strength of the stretch and torsional stress. The optimal cyclic directions and loading were established after multiple virtual simulations. A 4-degree tilt away from the panel provides 2 mm stretch and 3 mm sliding of each panel beneath the chambers, resulting a in 10 degree of rotation tilt of the chamber corners and torsional stretch on the PDMS chambers (Fig 3). The PDMS chambers are designed to be assembled on the MSS device after cell attachment has been confirmed. In addition, to determine the expected stretch load force on the internal surface of the chamber, a three-dimensional (3D) simulation program (Inventor, Autodesk Inc,

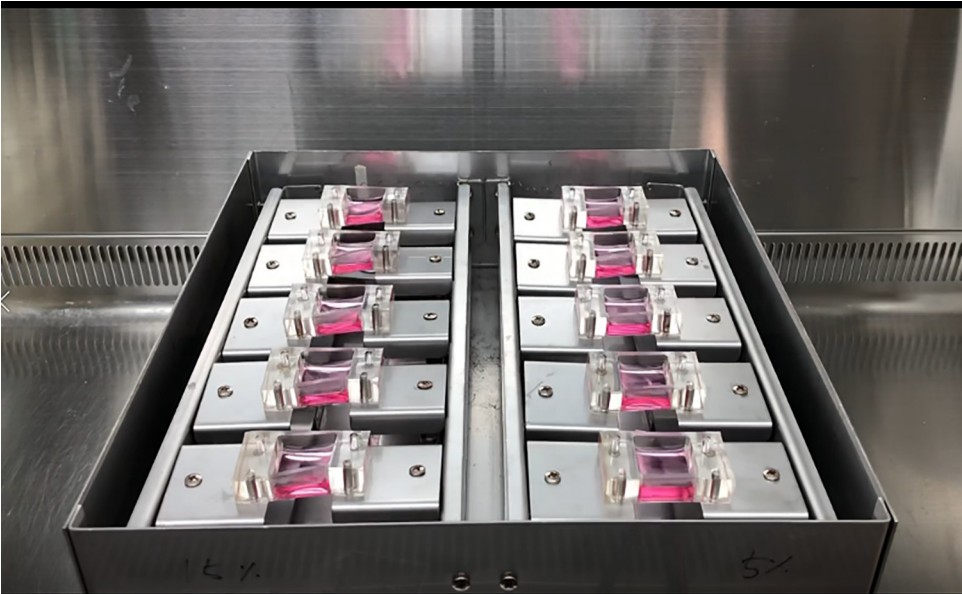

**Fig 1. Novel multi-torsional mechanical stretch stress loading device.** An open metal frame with multiple chambers seated parallel to the fixation panels.

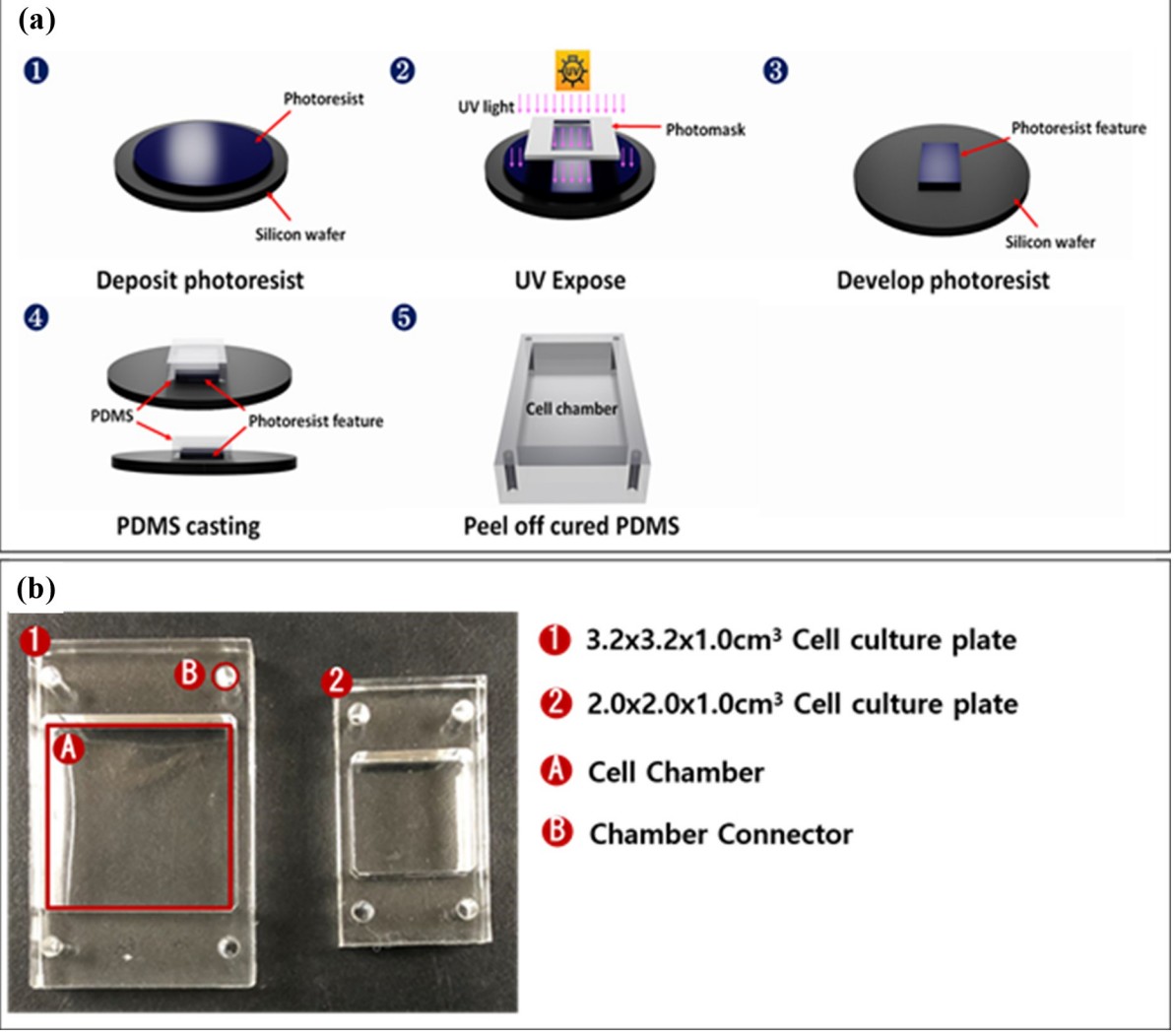

**Fig 2. Flexible polydimethylsiloxane chamber.** (a) Photolithography process; (b) PDMS stretch chambers compatible with the panels.

CA) was used. The torsional stress loaded on the external chamber surface was analyzed and presented as stress-strain ratios.

## Mechanical stretch stress loading on LF cells

LF cells were plated on the PDMS chamber at a density of $5.0 \times 10^4$ /mL. After 24 h of incubation, cell attachment to the cell chamber wall was verified, and the cells were subjected to MSS. Multi-torsional MSSs of 0% (no stretch–control), 5%, and 15% load were applied to multiple cell plate chambers simultaneously. The 5% and 15% MSS strength are relative strength compared to the maximal MSS that can be generated by the newly developed MSS load device. Three independent experiments were conducted.

## Lactate dehydrogenase assay

Lactate dehydrogenase (LDH), which is normally confined within the cell is released into culture supernatant after plasma membrane breakdown as result of cytotoxicity. This is a

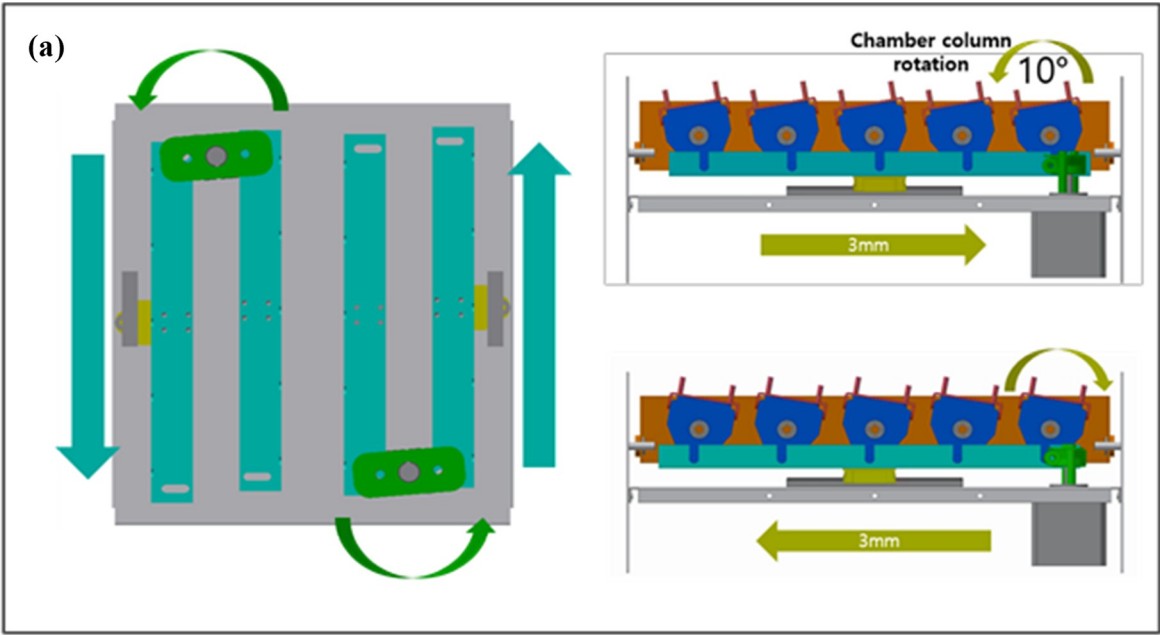

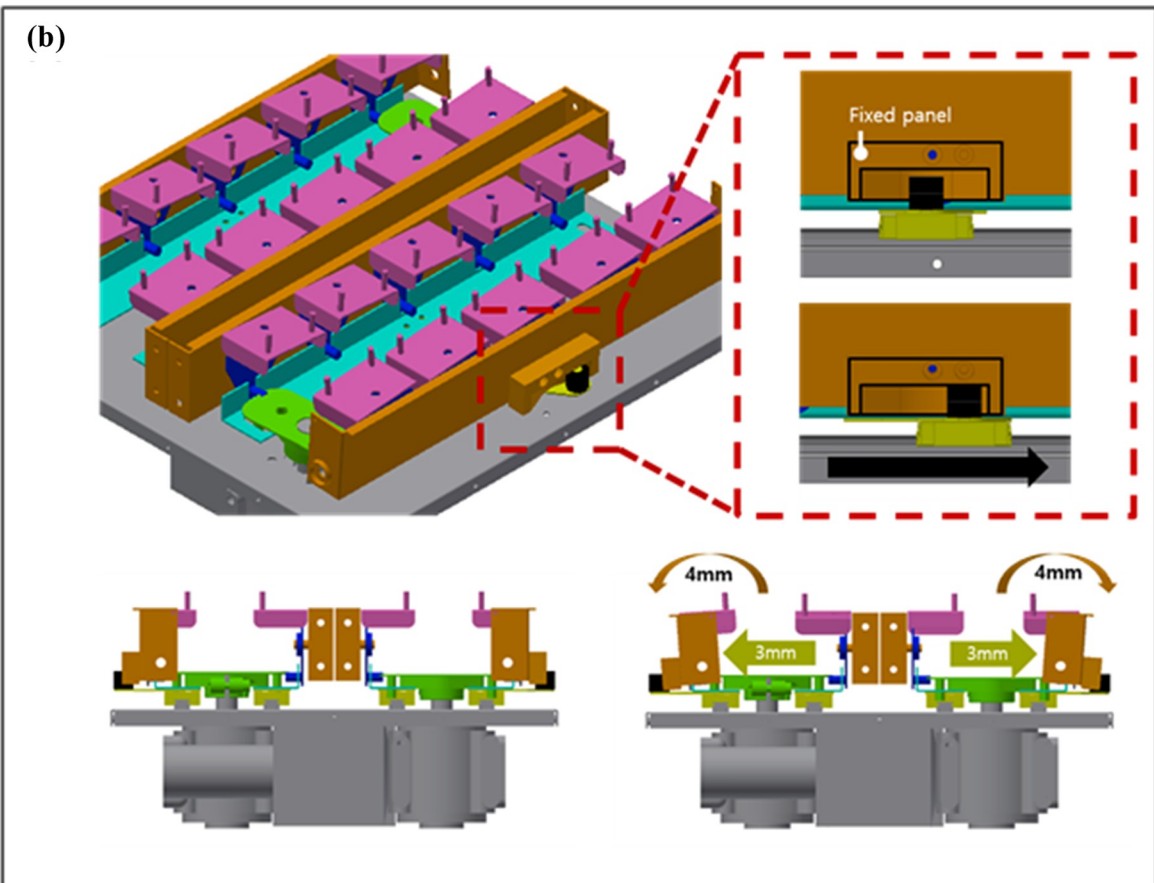

**Fig 3. Two-step motor generators.** (a) Producing a 10-degree rotational tilt of the chamber corners with resultant torsional stretch on the PDMS chambers (b) 4-degree tilt away from the panel results in 2 mm stretch and 3 mm sliding of each panel beneath the chambers.

commonly used methods allowing simultaneous analysis of specimens rapidly and cost-effectively [12]. LDH was assayed using LDH assay kit (Roche, Mannheim, Germany) after various MSS loading– 0%, 5% and 15%—at various durations– 24, 48, and 72 hours.

### Enzyme-linked immunosorbent assay

Concentrations of vascular growth factor (VEGF), interleukin (IL)-6 and IL-8, matrix metalloproteinase (MMP)-1 and MMP-3, and tissue inhibitor of metalloproteinase (TIMP)-1 and TIMP-2 were analyzed by ELISA using commercially available kits (R&D Systems, Minneapolis, MN) following the manufacturer's recommended protocols. All experiments were conducted in duplicate.

### Statistical analysis

Data are mean ± standard error (SE) for individual experiments using independent cell cultures. P values were calculated using Student's t-test or the Mann–Whitney U test, as appropriate according to sample size and distribution normality. P-values < 0.05 were considered to indicate statistical significance. All statistical analyses were performed using SPSS (version 20, SPSS, Chicago, IL, USA).

## Results

### Optimizing MSS load on LF cells

Using dual-step motor generators controlled by motor drivers, multi-torsional MSS was successfully loaded on the assembled cell chambers (S1 File). The tension-load was produced by optimizing the multi-torsional stretch strength and the cyclic load frequency, and 3D simulation was performed to visualize the expected load on the chambers (Fig 4). Morphologic

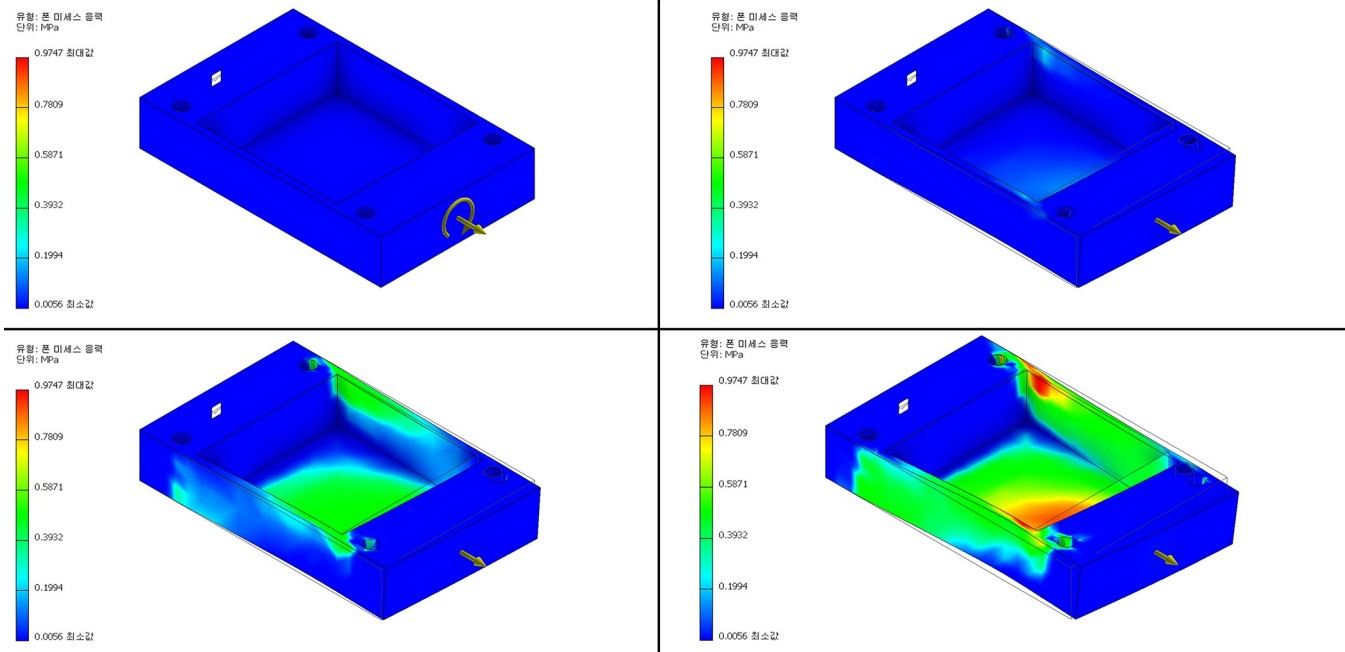

**Fig 4. Three-dimensional simulation of the expected load on the chambers.** The torsional stress loaded on the external chamber surface is shown as the stress-strain ratio.

evaluation of LF cells by optical microscopy revealed no significant phenotypic change after 24 h of 5% MSS, whereas cell death was noted with more prolonged MSS loading.

## Cytotoxicity assay of LF cells

LDH release from MSS loaded LF cells was measured to evaluate cytotoxicity at 24, 48, and 72 h after MSS loading. (Fig 5), MSS load on LF cells did not significantly affect LDH release at 5% stretch for 24 h, but at 15% stretch and exposure for 48 and 72 h LDH levels were significantly increased, indicating a cytotoxic effect. The ligamentum flavum cells did not present any phenotypic change under 5% stretch for 24h, but any further exposure to 5% MSS or any force stronger than 5% resulted in cell deaths. (Fig 6)

## Effect of MSS loading on inflammatory cytokines and vascular growth factors

IL-6 and IL-8 release after MSS loading was 296.80±89.35 and 72.27±11.12 ng/mL, respectively, significantly higher than in the control group (174.97±58.12 and 56.43±5.59 ng/mL, respectively). Furthermore, a significant increase in VEGF level following MSS loading (141.80 ±19.45 ng/mL) was observed compared to the control group (23.97±8.16 ng/mL) (Fig 7, Table 1).

## Effect of MSS load on production of ECM-regulating factors

MMP-1, MMP-3, MMP-9, TIMP-1, and TIMP-2 release levels from LF cells loaded with MSS at 5% for 24 h were measured to assess ECM remodeling. Following MSS loading, MMP-1, MMP-3, and MMP-9 release levels were 463.94±53.08, 579.92±90.43, and 25.77±1.84 ng/mL, respectively. The TIMP-2 release level was 320.00±16.34 ng/mL; TIMP-1 was undetectable. The MMP-1, MMP-3, and TIMP-2 release levels were significantly increased by MSS loading compared to the control group (330.15±35.41, 420.25±45.66, and 273.87±16.40 ng/mL, respectively) (Fig 7, Table 1).

## Discussion

LSCS is of interest to spinal physicians due to its increasing prevalence and clinical significance. Prior reports have indicated that LFH and LSCS are significantly associated with various clinical symptoms including back pain and radiculopathy with or without neurogenic claudication [9, 13–17]. The LF is a ligamentous structure lying over the dorsal aspect of the central

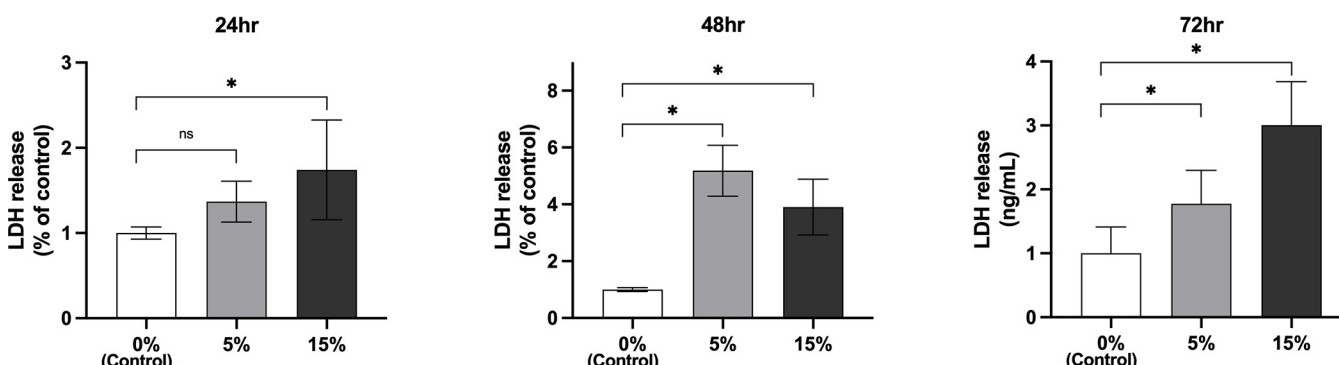

**Fig 5. Cytotoxicity assay of ligamentum flavum cells.** MSS loading at 5% stretch for 24 h on ligamentum flavum cells did not significantly affect the LDH release level. Stronger stretch (15%) and stretching for 48 h resulted in significantly increased LDH levels, indicating a cytotoxic effect.

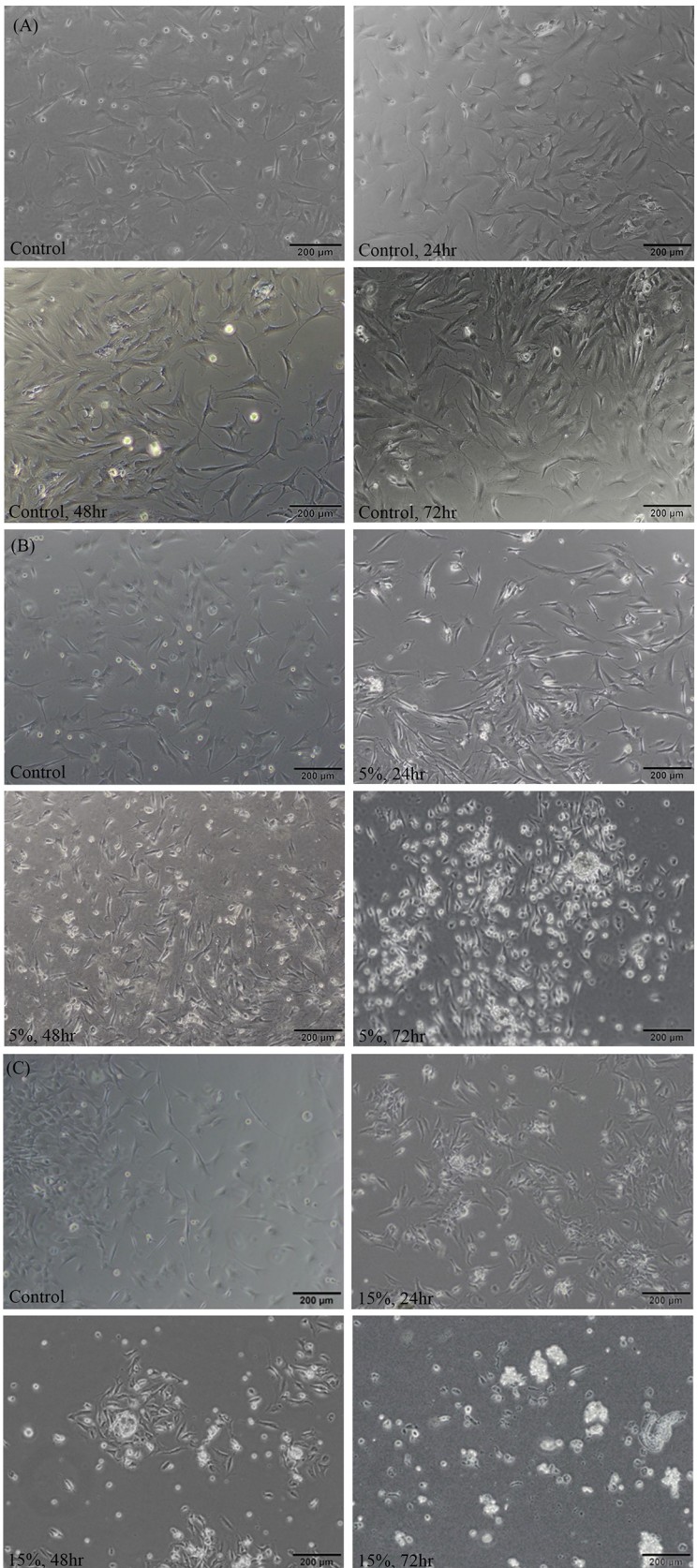

**Fig 6. Light microscopic images of ligamentum flavum cells.** Representative phase contrast light microscopic images of ligamentum flavum cells under different mechanical stretch stress (MSS) forces and different time durations. (A) Control MSS strength, (B) 5% MSS strength and (C) 15% MSS strength.

spinal canal, LFH directly results in physical narrowing of the spinal canal leading to clinical LSCS. To discover a novel therapeutic candidate targeting LFH, it is important to understand the pathomechanism of LFH. Elucidating the role of mechanical stress on the LF is critical, and inflammation/angiogenesis of the LF following mechanical stress are hallmarks of LFH.

We previously reported that inflammation and subsequent angiogenesis are involved in the pathomechanism of LFH *in vitro*, indicative of close relationships among inflammation, angiogenesis, and LFH [6]. In a follow-up study of the association between *in vitro* and clinical data, we discovered links among mechanical stress, angiogenesis, and LFH [7]. However, these studies were limited in that mechanical stress was not loaded directly onto the LR cells. Instead, the effects of mechanical stress were evaluated indirectly based on radiological findings. In this study, we developed a novel mechanical stress loading device with multidirectional torsion that mimics the mechanical load on LF tissue *in vivo*. Rather than inducing inflammation by transforming growth factor-β1 (TGF- β1) or interleukin-1β (IL-1β), we used mechanical stress on the LF and believe it reflects the effects of mechanical stress on LFH.

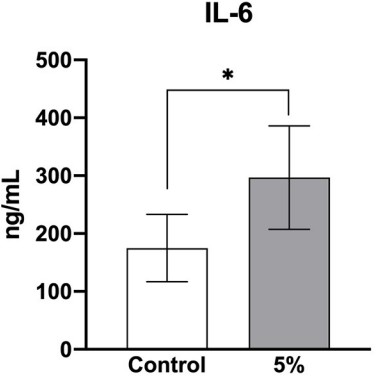
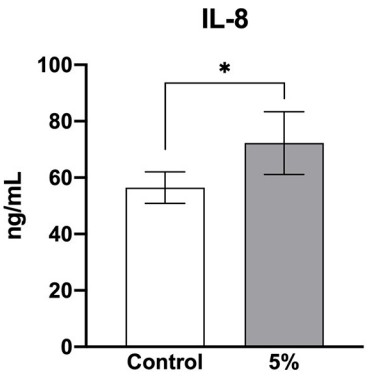
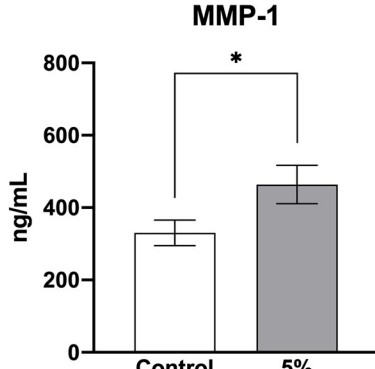

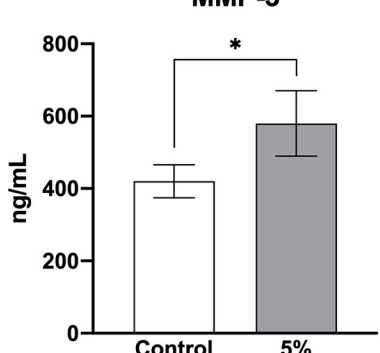
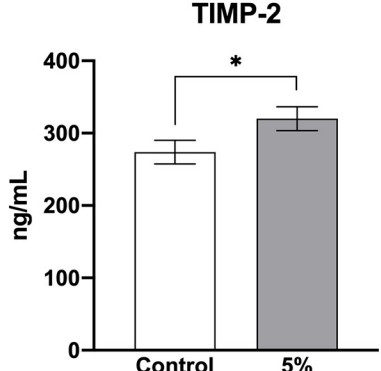
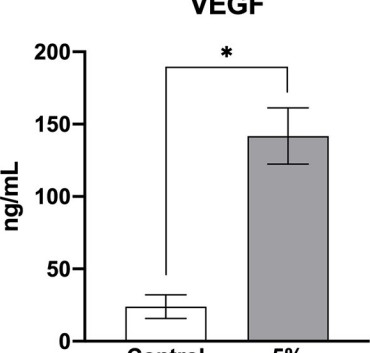

**Fig 7. Factors released in control versus 5% MSS.** IL-6, IL-8, VEGF, MMP-1, MMP-3, and TIMP-2 release from ligamentum flavum cells loaded with MSS at 5% for 24 h was significantly increased compared to the control group.

**Table 1. Inflammatory mediators, angiogenic factor, and ECM-regulating Molecule Production from LF cells.**

|  | Control | MSS load (5%, 24hrs) | p-value |
|---|---|---|---|
| **IL-6** | 174.97±58.12 | 296.80±89.35 | 0.013* |
| **IL-8** | 56.43±5.59 | 72.27±11.12 | 0.015* |
| **VEGF** | 23.97±8.16 | 141.80±19.45 | 0.028* |
| **MMP-1** | 330.15±35.41 | 463.94±53.08 | 0.049* |
| **MMP-3** | 420.25±45.66 | 579.92±90.43 | 0.024* |
| **MMP-9** | 35.46±4.82 | 25.77±1.84 | 0.009** |
| **TIMP-1** | NC | NC | - |
| **TIMP-2** | 273.87±16.40 | 320.00±16.34 | 0.049* |

ECM, extracellularmatrix; LF,ligamentum flavum; IL, interleukin; VEGF, vascular endothelial growth factor; MMP, matrix metalloproteinases; TIMP, tissue inhibitors of metalloproteinase; NC, Not Competent

Values unit; ng/mL. ± SE

* P<0.05, significant increase in concentration.

**P<0.05, significant decrease in concentration.

Our results demonstrated that multi-torsional MSS load for 24 h under 5% stretch force stimulation resulted in an increase in IL-6 and VEGF levels. IL-6 activates neutrophils, whose adhesion and fibrosis are promoted by increased expression of ECM-regulating molecules or cytokines [18]. A similar response leads to LFH after triggering inflammation in LF cells [6]. An increase in IL-6 can also upregulate mRNA expression and DNA synthesis of LF cells, resulting in ossification or fibrosis [19]. Our finding of a significant increase in IL-6 expression confirms that MSS loading induced inflammation in LF cells, mimicking the initial inflammatory phase of LFH. Likewise, VEGF initiates and stimulates the angiogenic cascade of LFH, and its concentration in degenerated or hypertrophied LF is significantly higher than that in healthy ligaments [6, 7, 10, 20]. MSS stimulation for 24 h resulted in marked elevation of VEGF expression in LF cells, indicating that MSS loading mimics the angiogenic cascade that occurs after an inflammatory reaction.

As well as inflammation and angiogenesis, the resultant changes in ECM-modulating factors (such as the elastin to collagen ratio) are important. Our data on ECM-regulating enzymes provide insight into the response of LF cells to mechanical stress. MMPs are endopeptidases involved in ECM homeostasis and in cell–cell interactions and angiogenesis. Significant changes or dysregulation of MMPs occur in cells during inflammation [21, 22], as well as in LF fibroblasts [8, 23]. MMP-1 is a collagenase for all collagen subtypes, and MMP-3 a broad-spectrum proteinase that regulates activation of other MMPs [24]. Elevated MMP-1 and MMP-3 levels after MSS stimulation by our novel multi-torsional stress loading device are compatible with prior reports confirming an association with LFH [25, 26]. This suggests the key role of mechanical stress in LFH as a result of altered ECM regulation in LF cells, indicating the dysregulation of regenerative potential and vulnerability to mechanical stress. However, Kim *et al.* reported increased expression of MMP-9 after inflammatory stimulation of LF cells *in vitro* [8], and Lakemeier *et al.* indicated that MMP-9 expression is higher in LFH tissue [27]. TIMPs also regulate ECM homeostasis, and TIMP-1 and TIMP-2 play key roles in fibrosis in various cell types by increasing proliferation. Park *et al.* hypothesized that TIMP-1 and TIMP-2 influence LFH by increasing ECM density and promoting hypertrophy by suppressing MMP activities [4]. This hypothesis was confirmed by the significant association between elevated TIMP-1 and TIMP-2 expression in LF fibroblasts and spinal stenosis, a reproducible finding of several different experiments of various methods [28]. This is compatible with our TIMP-1 and TIMP-2 expression data.

Mechanical stress is a key factor in LFH, as confirmed by *in vitro* [29–33], *in vivo* [10, 11], and clinical studies [7]. Chao *et al.* developed an *in vitro* method of loading stress on LF cells by centrifuging them in a horizontal microplate rotor [33]. Nakamura *et al.* loaded a cyclic uniaxial load to LF cells by attaching the cell culture chamber to a stretching apparatus [29], and Nakatani *et al.* loaded mechanical stress using a vacuum unit to pull a flexible cell culture plate from the center [31]. It is meaningful that centrifugal and cyclic one-dimensional mechanical forces on LF fibroblasts affected the mechanostress pathway. However, because one- and two-dimensional forces are unlike that on LF tissue *in vivo*, the accuracy of the model is unknown. Therefore, it is significant that we developed a reproducible repetitive mechanical stress loading device that recapitulates the mechanical stress on LF cells. The device will be used to provide insight into the role of direct mechanical stress on LFH *in vitro* and the cells' fate after mechanical stress loading.

In conclusion, we developed a novel multi-torsional mechanical stress loading device which can mimic the physiologic multi-dimensional mechanical stress that ligamentum flavum gets in vivo. Using this device, we confirmed that mechanical stress does enhance the inflammatory reaction in ligamentum flavum cells and enhances the production of angiogenic factors subsequently leading to possible LFH. ECM regulating enzymes' expression were also notably compatible with that of previous studies. We believe this development will help us understand the effect of MSS on LFH by providing a reproducible stress loading platform for ligamentum flavum in vitro experiments. Nevertheless, our study still has certain limitations which must be overcome by further future studies. We focused on the ligamentum flavum cellular protein production and expression and is currently lack of intracellular signaling RNA studies, and therefore cannot provide a comprehensive conclusion including subsequent cellular changes such as cytoskeletal remodeling or mechano-transduction signaling pathways. Further studies concerning intracellular signaling RNA with additional assay experiment, such as MTT assay, and various cell staining technique to validate cytoskeletal changes and cellular viability might lead us to a more information regarding the bio-mechanism. Another limitation is that we do not have in vivo results yet, and cannot have a concrete conclusion that our device perfectly mimics the actual multi-dimensional stretch stress on ligamentum flavum tissue. In order to do so, in vivo and further follow up experimental studies with more various loading stress are needed.

## Supporting information

**S1 File. Novel multi-torsional mechanical stretch stress loading device.** An open metal frame with multiple chambers seated parallel to the fixation panels. These chambers incorporate two-step motor generators producing rotational tilt and stretch forces.
(MP4)

## Author Contributions

**Conceptualization:** Woo-Keun Kwon, Chang Hwa Ham, Hyuk Choi, Seung Min Baek, Youn-Kwan Park, Hong Joo Moon, Woong Bae Park, Joo Han Kim.

**Data curation:** Woo-Keun Kwon, Chang Hwa Ham, Hyuk Choi, Seung Min Baek, Jae Won Lee, Woong Bae Park, Joo Han Kim.

**Formal analysis:** Woo-Keun Kwon, Chang Hwa Ham, Hyuk Choi, Seung Min Baek, Jae Won Lee, Youn-Kwan Park, Hong Joo Moon, Woong Bae Park, Joo Han Kim.

**Funding acquisition:** Joo Han Kim.

**Investigation:** Woo-Keun Kwon, Chang Hwa Ham, Hyuk Choi, Seung Min Baek, Jae Won Lee, Youn-Kwan Park, Hong Joo Moon, Woong Bae Park, Joo Han Kim.

**Methodology:** Woo-Keun Kwon, Chang Hwa Ham, Hyuk Choi, Seung Min Baek, Jae Won Lee, Youn-Kwan Park, Hong Joo Moon, Woong Bae Park, Joo Han Kim.

**Project administration:** Woo-Keun Kwon, Chang Hwa Ham, Hyuk Choi, Seung Min Baek, Joo Han Kim.

**Resources:** Woo-Keun Kwon, Chang Hwa Ham, Hyuk Choi, Seung Min Baek, Jae Won Lee, Youn-Kwan Park, Hong Joo Moon, Woong Bae Park, Joo Han Kim.

**Supervision:** Woo-Keun Kwon, Hyuk Choi, Youn-Kwan Park, Hong Joo Moon, Joo Han Kim.

**Validation:** Woo-Keun Kwon, Chang Hwa Ham, Hyuk Choi, Seung Min Baek, Jae Won Lee, Youn-Kwan Park, Hong Joo Moon, Woong Bae Park, Joo Han Kim.

**Visualization:** Chang Hwa Ham, Hyuk Choi, Seung Min Baek, Jae Won Lee, Joo Han Kim.

**Writing – original draft:** Woo-Keun Kwon, Chang Hwa Ham.

**Writing – review & editing:** Chang Hwa Ham.

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
