## [Decision Letter · Decision Letter 0]

21 Apr 2022

PONE-D-21-36196Elucidating the effect of mechanical stretch stress on the mechanism of ligamentum flavum hypertrophy: Development of a novel in vitro multi-torsional stretch loading device.PLOS ONE

Dear Dr. Joo Han Kim,

Thank you for submitting your manuscript to PLOS ONE. After careful consideration, we feel that it has merit but does not fully meet PLOS ONE’s publication criteria as it currently stands. Therefore, we invite you to submit a revised version of the manuscript that addresses the points raised during the review process.

We look forward to receiving your revised manuscript.

Kind regards,

Chuen-Mao Yang

Academic Editor

PLOS ONE

Journal Requirements:

Reviewers' comments:

Reviewer's Responses to Questions

**Comments to the Author**

1. Is the manuscript technically sound, and do the data support the conclusions?

Reviewer #1: Partly

Reviewer #2: Partly

2. Has the statistical analysis been performed appropriately and rigorously? 

Reviewer #1: Yes

Reviewer #2: I Don't Know

3. Have the authors made all data underlying the findings in their manuscript fully available?

Reviewer #1: Yes

Reviewer #2: Yes

4. Is the manuscript presented in an intelligible fashion and written in standard English?

Reviewer #1: Yes

Reviewer #2: No

5. Review Comments to the Author

Reviewer #1: The authors developed a novel multi-torsional mechanical stretch stress loading device for ligamentum flavum cells, and they evaluated its influence on ligamentum flavum hypertrophy in vitro. They revealed that the new device can recreate mechanical stress on these cells and enhance the production of inflammatory cytokines and angiogenic factors and alter the expression of ECM-regulating enzymes. This is a novel and interesting study. However, there are still some issues needed to be addressed in the present study.

1. Negative data should be demonstrated in Table 1.

2. Write the full name of MSS when it appears at the first time in the Abstract.

3. The phenotype of cell hypertrophy is an important concept in this research but there are no direct findings to clarify this pathology in the present study.

Reviewer #2: In this manuscript, a new device is designed to provide multi-torsional mechanical stretch stress to mimic the stress of ligamentum flavum druing lumbar spinal canal tenosis. It seems more suitable than centrifugal or cyclic modes. However, several points should be further clarified.

1. Why is the cell density of 1 X10^4 /mL used? It seems not growth confluence on the membrane of the device.

2. How is the percentage of stretch force determined? What's the meaning of 5%, or 15%? Does it mean that there is a setting of 100%? What's the scale?

3. Why are two kinds of statistical analysis for the p value? Why is different manner used?

4. In result section of cytotoxicity assay of LF cells, no 72 h of results shows. I am confused that the time duration of 24, 48 or 72 h. Does it mean that the cells are torsional stress for 24, 48 or even 72 h? Or cells are torsional stress for 5 or 10 min and rest for 24, 48 or 72h? It seems not reasonable to torsional stress cells for such long time duration.

5. It's a suggestion to test cell viability by one more assay experiment such as MTT assay.

6. The experiments of force stress with different strength loading should be tested.

7. Are the control group pictures of fig 3B and 3C the same? How long is the control duration? 0, 24, 48 or 72 h? In addition, a live and dead cell staining and a cytoskeleton change staining will be more suitable to prove the change of cell morphology.

8. The rationale of inlammation or ECM marker tested in this manuscript should be provided in introduction seciton.

9. the manuscript would be better to be revised by a naive speaker.

6. PLOS authors have the option to publish the peer review history of their article (what does this mean?). If published, this will include your full peer review and any attached files.

Reviewer #1: No

Reviewer #2: No

---

## [Author Response · Author response to Decision Letter 0]

26 May 2022

Answer to reviewers’ comments

Reviewer #1

1. Negative data should be demonstrated in Table 1

Thank you for the comment that needs to be revised. We have added the negative value of MMP-9 and TIMP to the Table 1, accordingly.

2. Write the full name of MSS when it appears at the first time in the Abstract.

Thank you for your comment. We have corrected the abstract accordingly to ‘Stretch strength of the device was optimized by applying 5% and 15% mechanical stretch stress (MSS) loads for 24, 48, and 72 h.’ [Line27,28]

3. The phenotype of cell hypertrophy is an important concept in this research but there are no direct findings to clarify this pathology in the present study.

Thank you for your comment that is worth discussing. While it would be also valuable to demonstrate the phenotypic characteristics at the cellular level, ligamentum hypertrophy occurs in constant and aberrant remodeling of extracellular matrix. The authors demonstrated the change in cytokines, enzymes, and growth factors that is associated with aberrant remodeling of ECM by applying stressful conditions to the LF cells that mimic in vivo mechanical stress.

Reviewer #2:

1. Why is the cell density of 1 X10^4 /mL used? It seems not growth confluence on the membrane of the device.

Thank you for your comment. We apologize for the mistyping in the manuscript. It is supposed to be 1x10^5/mL and we have corrected it accordingly as follows;

‘LF cells were plated on the PDMS chamber at a density of 5.0 � 104 /mL.’ [Line 127]

2. 2. How is the percentage of stretch force determined? What’s the meaning of 5%, or 15%? Does it mean that there is a setting of 100%? What’s the scale?

We greatly appreciate the reviewer for pointing out this ambiguity in explaining the definition of each MSS strength percentages. The 5% and 15% in this study is the strength compared to the possible maximal stretch force which can be generated by our new MSS device. So, it is the relative magnitude of MSS strength. As this is an in vitro study, we still cannot provide information regarding the relative strength compared to the in vivo physiologic mechanical stress. The setting of the magnitude of stretch strength is explained in Line 129-131

3. . Why are two kinds of statistical analysis for the p value? Why is different manner used?

Thank you for your comment which may have caused confusion. Two different statistical methods were used due to difference in normality distributions, which was conducted by Shapiro-Wilk test. In order to cause less confusion we corrected the corresponding sentence to ‘P values were calculated using Student’s t-test or the Mann–Whitney U test, as appropriate according to sample size and distribution normality determined by Shapiro-Wilk test. [Line 140-142]

4. In result section of cytotoxicity assay of LF cells, no 72 h of results shows. I am confused that the time duration of 24, 48 or 72 h. Does it mean that the cells are torsional stress for 24, 48 or even 72 h? Or cells are torsional stress for 5 or 10 min and rest for 24, 48 or 72h? It seems not reasonable to torsional stress cells for such long time duration.

Thank you for your comment. The authors have removed the 72hr stress from the figure, as 48hr stress already demonstrated the significant cytotoxicity. But we fully agree with the reviewer, that this may have created some confusions. Therefore, we added the 72hr stress to the figure. The purpose of this section of the study was to test the cellular viability of the experiment. Applying 5% stress for 24 hr seem not to cause cell death, as demonstrated by cytotoxic assay that does not (or insignificantly) differ from that of control. Therefore, we considered 24hr duration of stretch-torsional stress to be suitable for the subsequent experiment.

5. It’s a suggestion to test cell viability by one more assay experiment such as MTT assay.

Thank you for your comment. It is true that additional assay such as MTT may strengthen the cellular viability test. However, it is technically difficult at the moment as all experiments are already carried out. In the subsequent studies, the authors will consider carrying out the experiment. We have added following sentence in the discussion as part of limitation of this study; ‘Further studies concerning intracellular signaling RNA with additional assay experiment, such as MTT assay, and various cell staining technique to validate cytoskeletal changes and cellular viability might lead us to a more abundant conclusion regarding the bio-mechanism.’ [Line 273-275]

6. The experiments of force stress with different strength loading should be tested.

Thank you for your comment. Carrying out experiments with different loading stress would of course strengthen the data and the authors fully agree with your opinion. However, this is a pilot study to test out the novel device. In the future studies, along with additional viability test, multiple other loading stresses will be applied as suggested. We have added following sentence in the discussion as part of limitation of this study; ‘In order to do so, in vivo and further follow up experimental studies with more various loading stress are needed.’ [Line 277,278]

7. Are the control group pictures of fig 3B and 3C the same? How long is the control duration? 0, 24, 48 or 72 h? In addition, a live and dead cell staining and a cytoskeleton change staining will be more suitable to prove the change of cell morphology.

Thank you for your comment. There was a mistake during the image frame-working. We have corrected it with the appropriate image. We fully agree and appreciate with your opinion of staining the cells. However, at this stage of experiment it is not feasible to do so. We will use the cell staining technique in our follow-up studies We have added following sentence in the discussion as part of limitation of this study; ‘Further studies concerning intracellular signaling RNA with additional assay experiment, such as MTT assay, and various cell staining technique to validate cytoskeletal changes and cellular viability might lead us to a more abundant conclusion regarding the bio-mechanism.’ [Line 273-275]

8. The rationale of inlammation or ECM marker tested in this manuscript should be provided in introduction seciton.

Thank you for your comment. The authors agree with your comment that it is worth mentioning the rationale behind the markers used in this study. We used the following literatures to support the rationale for markers used; [Moon et al., 2012, Spine, Hur et al., 2015, Neurosurgery, Kim et al., 2016, J Neurosurg Spine]. We have added following sentence to the introduction section; ‘We evaluated the molecular biological responses related to inflammation, angiogenesis, and extracellular matrix (ECM) regulation that were reported in previous studies of ligamentum flavum cells to various stress loads to identify the stress load that best mimics LFH.’ [Line 67,68]

9. the manuscript would be better to be revised by a naive speaker.

Thank you for your comment. It is true for us – second language speakers – to leave the manuscript for native speakers to revise. We have used ‘Textcheck’ to revise the manuscript. This is a professional English consulting site for editing documents for publication in international journals. The website and reference number are http://www.textcheck.com/text/page/times and 21052202, respectively.

---

## [Decision Letter · Decision Letter 1]

21 Jul 2022

PONE-D-21-36196R1Elucidating the effect of mechanical stretch stress on the mechanism of ligamentum flavum hypertrophy: Development of a novel in vitro multi-torsional stretch loading device.PLOS ONE

Dear Dr. Kim,

Thank you for submitting your manuscript to PLOS ONE. After careful consideration, we feel that it has merit but does not fully meet PLOS ONE’s publication criteria as it currently stands. Therefore, we invite you to submit a revised version of the manuscript that addresses the points raised during the review process.

Three new reviewers have assessed the manuscript. They felt that some additional clarification is needed. Please see their detailed comments below.

We look forward to receiving your revised manuscript.

Kind regards,

Hanna Landenmark

Staff Editor

PLOS ONE

Journal Requirements:

Reviewers' comments:

Reviewer's Responses to Questions

**Comments to the Author**

1. If the authors have adequately addressed your comments raised in a previous round of review and you feel that this manuscript is now acceptable for publication, you may indicate that here to bypass the “Comments to the Author” section, enter your conflict of interest statement in the “Confidential to Editor” section, and submit your "Accept" recommendation.

Reviewer #1: All comments have been addressed

Reviewer #2: All comments have been addressed

Reviewer #3: All comments have been addressed

Reviewer #4: (No Response)

Reviewer #5: (No Response)

2. Is the manuscript technically sound, and do the data support the conclusions?

Reviewer #1: (No Response)

Reviewer #2: Yes

Reviewer #3: Partly

Reviewer #4: (No Response)

Reviewer #5: Partly

3. Has the statistical analysis been performed appropriately and rigorously? 

Reviewer #1: (No Response)

Reviewer #2: Yes

Reviewer #3: Yes

Reviewer #4: (No Response)

Reviewer #5: Yes

4. Have the authors made all data underlying the findings in their manuscript fully available?

Reviewer #1: (No Response)

Reviewer #2: Yes

Reviewer #3: Yes

Reviewer #4: (No Response)

Reviewer #5: Yes

5. Is the manuscript presented in an intelligible fashion and written in standard English?

Reviewer #1: (No Response)

Reviewer #2: Yes

Reviewer #3: Yes

Reviewer #4: (No Response)

Reviewer #5: Yes

6. Review Comments to the Author

Reviewer #1: (No Response)

Reviewer #2: This is an interesting work in introducing the new equipment and all my concerns are met. Looking forward to future application studies.

Reviewer #3: 1.As the experimental results, it can be seen that the device is effective, because it is a new method, we suggest that such a method of torsion and stretching should be compared with the more common FlexCell as possible.

2.In vitro part, Fig. 6A is confused, why with the increase of time (0, 24, 48, 72h), there is no corresponding increase in the degree of confluence of LF cells.

3.The cytotoxicity/viability part should be proved by better and more commonly used methods, maybe due to some limitations, the method of lactate dehydrogenase release have been used to evaluate cytotoxicity, the cell biological effect of LDH should be explained.

4.Mechanical stress is a key factor in LFH, In the in vitro part of the study of HLF, LF cells exhibits corresponding cellular phenotypic changes, such as increasing collagen I, collagen III and α-SMA, and there is no direct evidence for this in the author's paper, which does not seem to correspond to the HLF mentioned in the full text.

Reviewer #4: The authors developed a multi-torsional mechanical stretch stress loading device and intended to evaluate its influence on the development of ligamentum flavum hypertrophy. The results showed that mechanical stress enhances the production of inflammatory cytokines and angiogenic factors.

1. Line 186. Values unit ng/mL ± SE, which conflicts with the statement at line 140 in statistical analysis section. Need clarifications what were reported and please use consistent format throughout the manuscript.

Reviewer #5: In the manuscript, “Elucidating the effect of mechanical stretch stress on the mechanism of ligamentum flavum hypertrophy: Development of a novel in vitro multi-torsional stretch loadingdevice”, by Kwon et al., the authors indicated that mechanical stress enhanced the production of inflammatory cytokines and angiogenic factors, and altered the expression of ECM-regulating enzymes, possibly triggering ligamentum flavum hypertrophy. Several major concerns (listed below) greatly reduce the significance of the work.

1. Why test LDH to react to cytotoxicity? CCK-8 should be used to reflect cellular activity.

2. LDH is released most at 15% stretch with 24h and 72h treatments. What explains the lower release at 15% stretch than at 5% stretch after 48 hours?

3. Figure 6B shows that 5% stretch treatment after 72 h is better than 48 h in terms of cell status, contradicting the authors' results.

7. PLOS authors have the option to publish the peer review history of their article (what does this mean?). If published, this will include your full peer review and any attached files.

Reviewer #1: No

Reviewer #2: No

Reviewer #3: No

Reviewer #4: No

Reviewer #5: No

---

## [Author Response · Author response to Decision Letter 1]

22 Aug 2022

Answer to reviewers’ comments

Reviewer #1: (No Response)

Reviewer #2: This is an interesting work in introducing the new equipment and all my concerns are met. Looking forward to future application studies.

Thank you for taking your time to review this article and thank you for your positive comment, we appreciate it.

Reviewer #3: 

1.As the experimental results, it can be seen that the device is effective, because it is a new method, we suggest that such a method of torsion and stretching should be compared with the more common FlexCell as possible.

Thank you for your comment. Our novel device provides three dimensional torsion/stress to the cell layers, which we believe is the key advantage in terms of mimicking the physiologic loading to LF cell in vivo. However, we agree to the reviewer’s idea that comparing the results from a widely accepted method can be useful in terms of reproducibility, therefore in future studies, once this device gains acceptance, the we will do compare the effectiveness of the device as recommended. 

2.In vitro part, Fig. 6A is confused, why with the increase of time (0, 24, 48, 72h), there is no corresponding increase in the degree of confluence of LF cells.

Thank you for your comment. We believe the presented image shows significant difference of confluence by time difference, however as the reviewer feels differently, we have amended this figure by presenting an image from another duplicate experiment. We have changed the figure 6 as a whole, including 6A 72h, to be more representative and avoid any confusion.

3.The cytotoxicity/viability part should be proved by better and more commonly used methods, maybe due to some limitations, the method of lactate dehydrogenase release have been used to evaluate cytotoxicity, the cell biological effect of LDH should be explained.

Thank you for your comment. We agree with the reviewer’s comment as additional various cell viability testing would have resulted to a more fruitful result. However, this study was a study of a novel device, therefore we have started with a method that is rather familiar, reproducible and reliable, as suggested in previous peer-reviewed study (We added the references as below). We also agree with the reviewer that the manuscript needs the explanation on the background of cell biological effect of LDH. Therefore, we have added a section in materials and methods for the additional information as follows.

Added sentence: -

Lactate Dehydrogenase Assay

Lactate dehydrogenase (LDH), which is normally confined within the cell is released into culture supernatant after plasma membrane breakdown as result of cytotoxicity. This is a commonly used methods allowing simultaneous analysis of specimens rapidly and cost-effectively.(12) LDH was assayed using LDH assay kit (Roche, Mannheim, Germany) after various MSS loading – 0%, 5% and 15% - at various durations – 24, 48, and 72 hours. 

Added reference: -

12. Galluzzi L, Aaronson SA, Abrams J, Alnemri ES, Andrews DW, Baehrecke EH, et al. Guidelines for the use and interpretation of assays for monitoring cell death in higher eukaryotes. Cell Death & Differentiation. 2009;16(8):1093-107.

References advocated: -

Shin J, Hwang M, Back S, Nam H, Yoo C, Park J, et al. Electrical impulse effects on degenerative human annulus fibrosus model to reduce disc pain using micro-electrical impulse-on-a-chip. Scientific Reports. 2019;9(1):5827.

Hwang MH, Lee JW, Son H-G, Kim J, Choi H. Effects of photobiomodulation on annulus fibrosus cells derived from degenerative disc disease patients exposed to microvascular endothelial cells conditioned medium. Scientific Reports. 2020;10(1):9655

4.Mechanical stress is a key factor in LFH, In the in vitro part of the study of HLF, LF cells exhibits corresponding cellular phenotypic changes, such as increasing collagen I, collagen III and α-SMA, and there is no direct evidence for this in the author's paper, which does not seem to correspond to the HLF mentioned in the full text.

Thank you for your comment. We agree with the comment that for the direct evidence of the LFH, the full spectrum of cellular phenotypic changes, as suggested, as well as RNA expressions and intracellular signaling should be evaluated in a stepwise manner. However, in our previous study - Hur et al. Neurosurgery. 2015 and Kim et al. 2016 J Neurosurg Spine. 2016 – we discovered that VEGF and other angiogenesis, ECS regulation related signaling molecules and enzymes are related to LFH, acutally playing a role as the key initial step. As the former study is a comparison between LFH and non-LFH based on phenotypical characteristic, we proceeded with this study based on the target molecules from the former studies. We do believe that the current results represent the possibility of LFH in its current form, however, as the reviewer comments our future studies will include full spectrum of biologic assays describing the LFH.

The authors are aware of this limitation and therefore had included the following sentence in the discussion part; ‘We focused on the ligamentum flavum cellular protein production and expression and is currently lack of intracellular signaling RNA studies, and therefore cannot provide a comprehensive conclusion including subsequent cellular changes such as cytoskeletal remodeling or mechano-transduction signaling pathways. Further studies concerning intracellular signaling assay, might lead us to a more information regarding the bio-mechanism of LFH.’[Line 272-278]

Reviewer #4: The authors developed a multi-torsional mechanical stretch stress loading device and intended to evaluate its influence on the development of ligamentum flavum hypertrophy. The results showed that mechanical stress enhances the production of inflammatory cytokines and angiogenic factors.

Thank you for your comment

1. Line 186. Values unit ng/mL ± SE, which conflicts with the statement at line 140 in statistical analysis section. Need clarifications what were reported and please use consistent format throughout the manuscript.

Thank you for pointing out this in your comment. As pointed out, we have corrected the sentence from ‘Data are means ± standard deviations (SDs)’ to ‘Data are mean ± standard error (SE)’

Reviewer #5: In the manuscript, “Elucidating the effect of mechanical stretch stress on the mechanism of ligamentum flavum hypertrophy: Development of a novel in vitro multi-torsional stretch loading device”, by Kwon et al., the authors indicated that mechanical stress enhanced the production of inflammatory cytokines and angiogenic factors, and altered the expression of ECM-regulating enzymes, possibly triggering ligamentum flavum hypertrophy. Several major concerns (listed below) greatly reduce the significance of the work.

1. Why test LDH to react to cytotoxicity? CCK-8 should be used to reflect cellular activity.

Thank you for your comment. We agree with the reviewer’s comment as additional cell viability testing, such as CCK-8, would give more strength to the study result. However, this study was a study of a novel device, therefore we have started with a method that is rather familiar, reproducible and reliable, as suggested in previous peer-reviewed study. 

The authors are aware of this limitation and therefore had included the following sentence in the discussion part; ‘We focused on the ligamentum flavum cellular protein production and expression and is currently lack of intracellular signaling RNA studies, and therefore cannot provide a comprehensive conclusion including subsequent cellular changes such as cytoskeletal remodeling or mechano-transduction signaling pathways. Further studies concerning intracellular signaling RNA with additional assay experiment, such as MTT assay, and various cell staining technique to validate cytoskeletal changes and cellular viability might lead us to a more abundant conclusion regarding the bio-mechanism.’ [Line 272-278]

2. LDH is released most at 15% stretch with 24h and 72h treatments. What explains the lower release at 15% stretch than at 5% stretch after 48 hours?

Thank you for your comment. Although, 5% stretch after 48 hours seem higher release than 15%, it is statistically insignificant. Given that the sample size is small it is unlikely that this reversed trend of LDH is meaningful. Another possible interpretation could be that the cell cytotoxicity is already present at a certain time point (after 48 hours) and further cell stress does not necessarily lead to a proportional increase of LDH release afterwards. However, we focused on statistically significant results, which correlated to the purpose of the experiment.

3. Figure 6B shows that 5% stretch treatment after 72 h is better than 48 h in terms of cell status, contradicting the authors' results.

Thank you for your comment. We believe the presented image shows significant difference of confluence by time difference, however as the reviewer feels differently, we have amended this figure by presenting an image from another duplicate experiment. We have changed the figure 6 as a whole, including 6A 72h, to be more representative and avoid any confusion.

---

## [Decision Letter · Decision Letter 2]

12 Sep 2022

Elucidating the effect of mechanical stretch stress on the mechanism of ligamentum flavum hypertrophy: Development of a novel in vitro multi-torsional stretch loading device.

PONE-D-21-36196R2

Dear Dr. Kim,

We’re pleased to inform you that your manuscript has been judged scientifically suitable for publication and will be formally accepted for publication once it meets all outstanding technical requirements.

Kind regards,

Dragana Nikitovic, Ph.D

Academic Editor

PLOS ONE

Additional Editor Comments (optional):

Reviewers' comments:

Reviewer's Responses to Questions

**Comments to the Author**

1. If the authors have adequately addressed your comments raised in a previous round of review and you feel that this manuscript is now acceptable for publication, you may indicate that here to bypass the “Comments to the Author” section, enter your conflict of interest statement in the “Confidential to Editor” section, and submit your "Accept" recommendation.

Reviewer #1: All comments have been addressed

Reviewer #4: (No Response)

2. Is the manuscript technically sound, and do the data support the conclusions?

Reviewer #1: Yes

Reviewer #4: (No Response)

3. Has the statistical analysis been performed appropriately and rigorously? 

Reviewer #1: Yes

Reviewer #4: (No Response)

4. Have the authors made all data underlying the findings in their manuscript fully available?

Reviewer #1: Yes

Reviewer #4: (No Response)

5. Is the manuscript presented in an intelligible fashion and written in standard English?

Reviewer #1: Yes

Reviewer #4: (No Response)

6. Review Comments to the Author

Reviewer #1: (No Response)

Reviewer #4: (No Response)

7. PLOS authors have the option to publish the peer review history of their article (what does this mean?). If published, this will include your full peer review and any attached files.

Reviewer #1: No

Reviewer #4: No

---

## [Editor Report · Acceptance letter]

6 Oct 2022

PONE-D-21-36196R2 

Elucidating the effect of mechanical stretch stress on the mechanism of ligamentum flavum hypertrophy: Development of a novel *in vitro* multi-torsional stretch loading device. 

Dear Dr. Kim:

I'm pleased to inform you that your manuscript has been deemed suitable for publication in PLOS ONE. Congratulations! Your manuscript is now with our production department. 

Kind regards, 

on behalf of

Dr. Dragana Nikitovic 

Academic Editor

PLOS ONE